# Discriminatory attitude towards people living with HIV/AIDS and its associated factors among adult population in 15 sub-Saharan African nations

Achamyeleh Birhanu Teshale [ID]*, Getayeneh Antehunegn Tesema

Department of Epidemiology and Biostatistics, Institute of Public Health, College of Medicine and Health Sciences, University of Gondar, Gondar, Ethiopia

* achambir08@gmail.com

## Abstract

### Background

Discrimination of people living with HIV/AIDS is one of the reported obstacles to the achievement of universal access to HIV/AIDS prevention, treatment, care, and support programs. Many international agencies have made combating HIV/AIDS stigma and discrimination a top priority. However, previous evidence in different parts of Africa revealed that the magnitude of HIV/AIDS-related discriminatory attitude is significantly high.

### Objective

To assess discriminatory attitude towards people living with HIV/AIDS and its associated factors among the adult population in 15 sub-Saharan African nations.

### Methods

We have used the 15 Demographic and Health Survey data that were conducted in sub-Saharan Africa (SSA) from 2015 to 2019/20. Each country's data was appended and a total weighted sample of 318,186 (unweighted sample = 315,448) adults who had ever heard of AIDS was used for the final analysis. The two discriminatory attitude questions were used to get the outcome variable and those who answered "Yes" or "don't know" for both questions were counted as if they had no discriminatory attitude towards people living with HIV/AIDS. To assess the factors associated with discriminatory attitude towards people living with HIV/AIDS, we have fitted a multilevel binary logistic regression model. Bivariable analysis was done to select eligible variables for the multivariable analysis. Finally, variables with p<0.05, in the multivariable analysis, were considered as significant predictors of discriminatory attitude towards people living with HIV/AIDS.

### Results

The prevalence of discriminatory attitude towards HIV/AIDS in the 15 sub-Saharan African nations was 47.08% (95% CI: 47.08, 47.42), which ranges from 17.64% (95% CI: 17.22,

**Data Availability Statement:** All relevant data are within the manuscript and Supporting information.

**Funding:** The author(s) received no specific funding for this work.

**Competing interests:** The authors have declared that no competing interests exist.

**Abbreviations:** AIDS, Acquired Immunodeficiency Virus; AOR, Adjusted Odds Ratio; CI, Confidence Interval; DHS, Demographic and Health Surveys; HIV, Human Immunodeficiency Virus; ICC, Intraclass Correlation Coefficient; PCV, Proportionate Change in Variance; SSA, Sub-Saharan Africa.

18.07) in Malawi to 79.75% (95% CI: 79.02, 80.45) in Guinea. In the multivariable analysis, both individual level and community level variables were significantly associated with discriminatory attitude towards people living with HIV/AIDS. Being younger age, no formal education, never married, low socioeconomic status, male-headed household, non-contraceptive use, no mass media exposure, and incorrect comprehensive knowledge towards HIV/AIDS were among the individual-level factors that were associated with higher odds of discriminatory attitude towards people living with HIV/AIDS. While being from urban residence and the western SSA region were among the community-level factors that were significantly associated with higher odds discriminatory attitude towards people living with HIV/AIDS.

## Conclusion

The prevalence of discriminatory attitude towards people living with HIV/AIDS in 15 sub-Saharan African nations was high. Both individual and community-level factors were associated with discriminatory attitude towards people living with HIV/AIDS. Therefore, special attention should be given to those who are poor, uneducated, and younger adults. In addition, it is better to strengthen the accessibilities of different media for adult populations to create an appropriate attitude towards people with HIV/AIDS.

## Background

The burden of the global HIV epidemic is disproportionately concentrated in sub-Saharan Africa, in which in 2017 about 75% of deaths and 65% of new infections occurred and where 71% of people living with HIV exist in [1, 2]. To stop and reverse the spread of HIV/AIDS, Sustainable Development Goal 3 calls for the end of the epidemic by 2030. In addition, the Joint United Nations Programme on HIV/AIDS (UNAIDS) sets a goal of reducing both new infections and deaths by 2030. Despite these goals, a recent review of the state of HIV concluded that the world is not at the potential to end the HIV epidemic [3–5].

Discrimination, one of the reported obstacles to the achievement of universal access to HIV/AIDS prevention, treatment, care, and support programs, is a differential action or behavior towards the stigmatized person based on those attitudes and perceptions [6]. The reason behind HIV/AIDS-related discrimination is due to the nature of HIV/AIDS, which is due to its incurability, and fatality, its contagiousness and transmissibility, and the repellent, ugly, and upsetting appearance of the infected individual at the advanced stages of the disease [7]. The other reason will be its transmission modality, transmitted through sexual intercourse that is viewed as a consequence of sexual immoral behaviors; thus, people living with HIV are severely discriminated regardless of how they actually became infected [8].

Many international agencies including the World Health Organization, the Joint United Nations Programme on HIV and AIDS, and the United States Agency for International Development have made combating HIV/AIDS stigma and discrimination a top priority, as this phenomenon undermines public health efforts to combat the pandemic [9, 10]. Extensive discriminatory attitude in a population can impair people's desire to be tested for HIV, their commencement of and adherence to antiretroviral therapy, social support, disclosing their status to family members, colleagues, and sexual partners and finally it affects their quality of life [11–16].

Previous evidence in different parts of Africa revealed that the magnitude of HIV/AIDS-related discriminatory attitude is significantly high, which ranges from 40% to 93.8% [17–19]. Studies have shown that different factors such as sex of respondent, educational status, mass media exposure, age, employment, comprehensive knowledge, place of residence, and community level of education are important factors associated with discriminatory attitude towards people living with HIV/AIDS [17, 20–28].

Due to the abovementioned negative effects of discriminatory attitude, there must be interventions to reduce discriminatory attitudes for combating HIV/AIDS transmission, as well as for increasing the quality of life of people living with HIV. Even though discrimination is a devastating issue, particularly in sub-Saharan Africa, the factors associated with discriminatory attitude towards people living with HIV/AIDS in sub-Saharan Africa is understudied. Therefore, this study aimed to assess discriminatory attitudes towards people living with HIV/AIDS and its associated factors in 15 sub-Saharan African nations. This could help policymakers and other responsible bodies combat this devastating problem at the country and regional levels.

## Methods

### Data source and study population

We have used DHS that were conducted from 2015 to 2019/20, conducted in the last five years. Even though, in SSA, there were 20 countries DHS that were conducted between 2015 and 2019/2020, for this study, only 15 countries DHS was used since there were not data about the outcome variable in the three country surveys (South Africa, Senegal, Chad, Tanzania, and Rwanda DHS had no observation regarding the outcome variable). First, we have appended both individual record women's file and men's file for each country. Finally, each countries data was appended and a total weighted sample of 318,186 (98,322 men and 219,864 women) adults, total unweighted sample of 315448 adults, who had ever heard of AIDS were used for the final analysis (Table 1).

### Study variables

**Outcome variable.**   Our outcome variable was the discriminatory attitude towards people living with HIV/AIDS, which is a binary outcome variable. The two discriminatory attitude questions (Would you buy vegetables from a vendor with HIV and children with HIV should be allowed to attend school with children without HIV) were used to get the outcome variable. Individuals had a discriminatory attitude towards people living with HIV/AIDS only if they answer "No" for both questions. Those who answered "Yes" or "don't know" had counted as if they had no discriminatory attitude towards people living with HIV/AIDS [29].

**Independent variables.**   For assessing the factors associated with discriminatory attitude towards people living with HIV/AIDS, both individual and community level independent variables were incorporated.

*Individual-level variables.* Sex, age, educational level, current marital status, wealth index, occupation, media exposure, sex of household head, contraceptive use, and comprehensive knowledge towards HIV/AIDS were incorporated as individual-level variables.

*Community-level variables.* Residence, African region, community illiteracy level, and community level of media exposure were the community-level variables.

**Operational definition.**   *Media exposure*. Created by combining whether a respondent reads a newspaper, listens to the radio, and watches television and coded as yes (if an individual had been exposed to at least one of these media) and no (otherwise).

*Comprehensive knowledge of HIV/AIDS*. A composite score of six different questions: 1. We can get HIV by witchcraft or supernatural means (yes/no), 2. Consistent use of condoms

**Table 1. Percentage distribution of study participants by country and region of Africa.**

| Countries with their Region | Year of study | Unweighted sample size (315,186) | Unweighted Percentage (%) | Weighted sample size (318,186) | Weighted Percentage (%) |
|---|---|---|---|---|---|
| **Eastern Africa** | | **145,385** | **46.09** | **145,797** | **45.82** |
| Burundi | 2016/17 | 23,070 | 7.31 | 23,050 | 7.24 |
| Ethiopia | 2016 | 25,542 | 8.10 | 25,927 | 8.15 |
| Uganda | 2018/19 | 23,438 | 7.43 | 23,437 | 7.37 |
| Zambia | 2018 | 24,361 | 7.72 | 24,463 | 7.69 |
| Zimbabwe | 2015 | 17,831 | 5.65 | 17,845 | 5.61 |
| Malawi | 2017 | 31,143 | 9.87 | 31,075 | 9.77 |
| **Western Africa** | | **134,846** | **42.75** | **136,418** | **42.87** |
| Benin | 2017/18 | 13,262 | 4.20 | 13,193 | 4.15 |
| Sierra Leone | 2019 | 20,498 | 6.50 | 20,719 | 6.51 |
| Gambia | 2019/20 | 15,627 | 4.95 | 15,740 | 4.95 |
| Guinea | 2018 | 11,957 | 3.79 | 12,112 | 3.82 |
| Liberia | 2019/20 | 11,251 | 3.57 | 11,293 | 3.55 |
| Mali | 2018 | 11,803 | 3.74 | 12,653 | 3.98 |
| Nigeria | 2018 | 50,448 | 15.99 | 50,708 | 15.94 |
| **Central Africa** | | **35,217** | **11.16** | **35,970** | **11.30** |
| Angola | 2015/16 | 16,161 | 5.12 | 16,736 | 5.26 |
| Cameroon | 2018 | 19,056 | 6.04 | 19,235 | 6.05 |

during sexual intercourse can reduce the chance of getting HIV (yes/no), 3. Having just one uninfected faithful partner can reduce the chance of getting HIV (yes/no), 4. Can get HIV from mosquito bites (Yes/no), 5. Can get HIV by sharing food with a person who has HIV/AIDS (Yes/no), and 6. A healthy-looking person can have HIV (Yes/no). Then the respondent had correct comprehensive knowledge if she/he answers all the six questions correctly and not knowledgeable if she/he did not give the correct answer for at least one of the questions.

*Community illiteracy level.* It was the proportion of adults with no formal education derived from data on respondent's level of education. Then, it was categorized using national median value to values: low (if the individual was from communities in which ≤50% of adults had no formal education) and high (if the individual was from communities in which >50% of adults had no formal education) community illiteracy level.

*Community-level of media exposure.* The proportions of adults who were exposed to media within a specific cluster. It was categorized in the same fashion as the community illiteracy level into low and high community-level of media exposure.

## Data management and statistical analysis

We have used Stata version 14 (StataCorp. 2015. Stata: Release 14. Statistical Software. College Station, TX: StataCorp LLC) for appending, extracting, and analyzing data. Throughout the statistical analysis, we have weighted the data to restore the representativeness of the sample and to get a robust standard error (an appropriate statistical estimate). The results of descriptive analyses were reported using texts, tables, and graphs.

To assess the factors that were associated with discriminatory attitude towards people living with HIV/AIDS, we have used a multilevel logistic regression analysis since the DHS data had hierarchical nature and the outcome variable was binary. We have fitted four models (Model 1, 2, 3, and 4) while conducting multilevel analysis. The model I was fitted with only the outcome variable to assess the variability of discriminatory attitude towards people living with

HIV/AIDS between clusters; Model 2 fitted with only individual-level variables; Model 3 and Model 4 was fitted with community level variables only and both individual and community level variables respectively.

To determine the community level variability of discriminatory attitude towards people living with HIV/AIDS, we have conducted a random-effects analysis. In the random effect analysis we have calculated the Intraclass Correlation Coefficient (ICC); to indicate the amount of variation of having a discriminatory attitude towards people living with HIV/AIDS that could be due to variability between cluster/communities and Proportional Change in Variance (PCV); to show to what extent discriminatory attitude towards people living with HIV/AIDS was explained by the fitted model. Of the fitted four models, the best-fitted model was selected using deviance (a model with the lowest deviance, model 4, was the best-fitted model).

For selecting eligible variables for the multivariable analysis, we have done the bivariable analysis. Those variables with a p<0.20, in the bivariable analysis, were eligible for the multi-variable analysis and, in the multivariable analysis; the adjusted odds ratio (AOR) with its 95% confidence interval (CI) was reported. Finally, variables with p≤0.05 were considered as significant predictors of discriminatory attitude towards people living with HIV/AIDS.

### Ethical consideration

Since we were using publicly accessible data, ethical approval was not needed. In addition, this research was considered exempt by the Institute of Public Health, College of Medicine and Health Sciences, University of Gondar Institutional Review Committee. However, by registering or online requesting we have accessed the data set from the DHS website (https://dhsprogram.com).

## Results

### Sociodemographic characteristics of respondents

Total weighted samples of 318,186 individuals were used for the final analysis. More than two-thirds (69.10%) of participants were females. The majority (60.16%) of respondents were married during the survey and around 24.92% of the respondents were from the richest households. Around three-fourth (76.96%) of the study participants were from male-headed households. A majority (70.81%) of respondents were exposed to at least one media (radio, television, or newspaper) and only 41.39% of respondents had comprehensive knowledge regarding HIV/AIDS (Table 2).

### Prevalence of discriminatory attitude towards people living with HIV/AIDS in 15 sub-Saharan African nations

The prevalence of discriminatory attitude towards people living with HIV/AIDS in SSA was 47.08% (95% CI: 47.08, 47.42), with huge variation between countries that ranges from 17.64% (95% CI: 17.22, 18.07) in Malawi to 79.75% (95% CI: 79.02, 80.45) in Guinea (Fig 1).

### Prevalence by African region, particularly by sub-Saharan African region

The prevalence of discriminatory attitude towards people living with HIV/AIDS was highest, 68.20 (95%CI: 67.95, 68.45), in the western African region (Fig 2).

**Table 2. Sociodemographic characteristics of respondents.**

| Variables | Weighted frequency (n = 318,186) | Percentage (%) |
|---|---|---|
| Sex | | |
| Male | 98322 | 30.90 |
| Female | 219864 | 69.10 |
| Age | | |
| 15–19 | 69750 | 21.92 |
| 20–24 | 57407 | 18.04 |
| 25–29 | 53737 | 16.89 |
| 30–34 | 44922 | 14.12 |
| 35–39 | 39120 | 12.29 |
| 40–44 | 29503 | 9.27 |
| 45–49 | 23747 | 7.46 |
| Educational status | | |
| No education | 80579 | 25.32 |
| Primary | 98201 | 30.86 |
| Secondary | 116107 | 36.49 |
| Higher | 23299 | 7.32 |
| Occupation | | |
| Not working | 103577 | 32.55 |
| Working | 214609 | 67.45 |
| Marital status | | |
| Single | 104807 | 32.94 |
| Married | 191411 | 60.16 |
| Widowed/separated/divorced | 21968 | 6.90 |
| Wealth status | | |
| Poorest | 52009 | 16.35 |
| Poorer | 56870 | 17.87 |
| Middle | 61092 | 19.20 |
| Richer | 68909 | 21.66 |
| Richest | 79306 | 24.92 |
| Sex of household head | | |
| Male | 232766 | 76.96 |
| Female | 69680 | 23.04 |
| Contraceptive use | | |
| No | 232676 | 73.13 |
| Yes | 85510 | 26.87 |
| Media exposure | | |
| No | 92875 | 29.19 |
| Yes | 225311 | 70.81 |
| Comprehensive knowledge of HIV/AIDS | | |
| No | 186498 | 58.61 |
| Yes | 131688 | 41.39 |
| Residence | | |
| Urban | 130125 | 40.90 |
| Rural | 188061 | 59.10 |
| Community-level of women literacy | | |
| Low | 165368 | 51.97 |
| High | 152818 | 48.03 |

(*Continued*)

**Table 2.** (Continued)

| Variables | Weighted frequency (n = 318,186) | Percentage (%) |
|---|---|---|
| Community-level media exposure | | |
| Low | 154531 | 48.57 |
| High | 163655 | 51.43 |

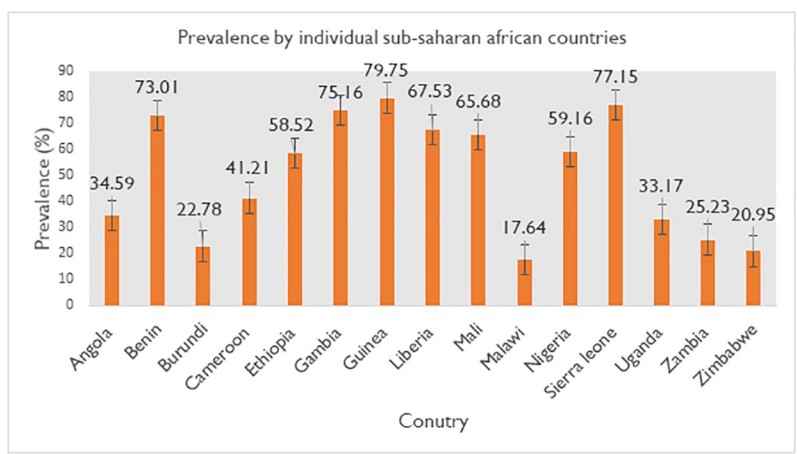

**Fig 1. Prevalence of discriminatory attitude towards people living with HIV/AIDS by countries in sub-Saharan Africa.**

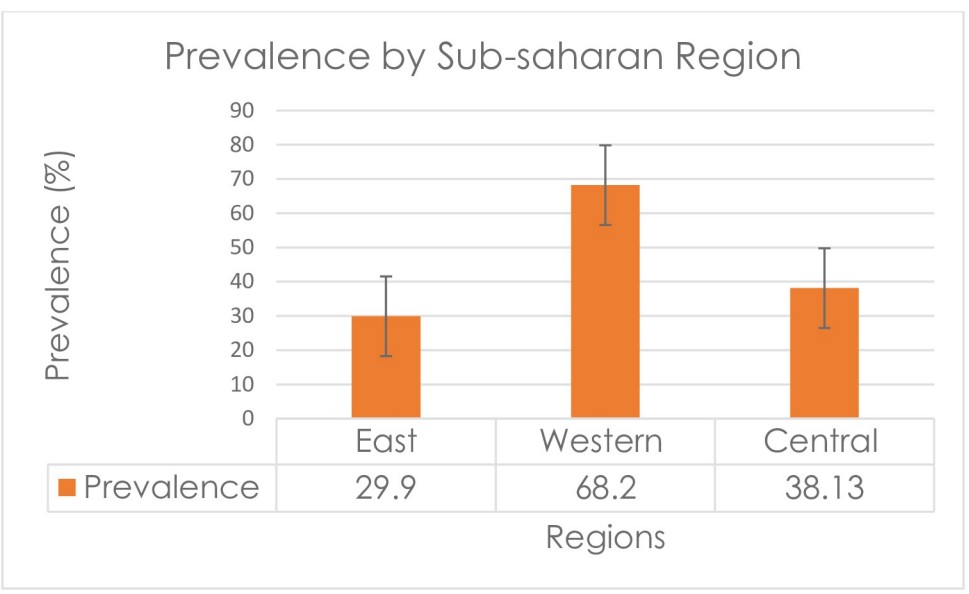

**Fig 2. Prevalence of discriminatory attitude towards people living with HIV/AIDS by sub-Saharan Africa regions.**

## Factors associated with discriminatory attitudes towards people living with HIV/AIDS

**Fixed effect analysis.** Both bivariable and multivariable analysis was conducted. In this study, all independent variables had p<0.20 in the bivariable analysis. Therefore, all variables were eligible for the multivariable analysis. In the multivariable analysis, both individual level and community level variables were associated with discriminatory attitude towards people living with HIV/AIDS. Age of the respondent, educational status, marital status, wealth index, sex of household head, contraceptive use, mass media exposure, and comprehensive knowledge towards HIV/AIDS were among the individual-level factors that were associated with discriminatory attitude towards people living with HIV/AIDS. While residence and SSA region were among the community-level factors that were significantly associated with discriminatory attitude towards people living with HIV/AIDS (Table 3).

The odds of having discriminatory attitude towards people living with HIV/AIDS among participants in the age group 20–24, 25–29, 30–34, 35–39, 40–44, and 45–49 years were 20% (AOR = 0.80; 95%CI: 0.78, 0.83), 24% (AOR = 0.76; 95%CI: 0.73, 0.79), 33% (AOR = 0.67; 95% CI: 0.64, 0.70), 33% (AOR = 0.67; 95%CI: 0.64, 0.70), 37% (AOR = 0.63; 95%CI: 0.60, 0.66), and 35% (AOR = 0.65; 95%CI: 0.62, 0.68) lower as compared to those participants who were in the age group 15–19 years respectively. Being having primary, secondary, and higher education had 56% (AOR = 0.44; 95%CI: 0.42, 0.46), 63% (AOR = 0.37; 95%CI: 0.36, 0.39), and 73% (AOR = 0.27; 95%CI: 0.26, 0.29) lower odds of discriminatory attitude towards people living with HIV/AIDS respectively, as compared to those who had no formal education. The odds of having a discriminatory attitude towards people living with HIV/AIDS among those who were married and divorced/separated/widowed were 7% (AOR = 0.93; 95%CI: 0.90, 0.96), and 30% (AOR = 0.70; 95%CI: 0.66, 0.74) lower respectively, as compared to those who were never married. Regarding wealth status, those individuals who were from poorer, middle, rich, and richest households had lower odds of having a discriminatory attitude towards people living with HIV/AIDS as compared to those who were from the poorest households. The odds of having a discriminatory attitude towards people living with HIV/AIDS among those individuals who were from female-headed households were 10% (AOR = 0.90; 95%CI: 0.87, 0.92) lower as compared to their counterparts. Being using contraceptive methods was associated with 26% (AOR = 0.74; 95%CI: 0.72, 0.76) lower odds of discriminatory attitude towards people living with HIV/AIDS as compared to their counterparts. Being having mass media exposure was associated with 5% (AOR = 0.95; 95%CI: 0.92, 0.98) lower odds of having a discriminatory attitude towards people living with HIV/AIDS as compared to those who had no mass media exposure. The odds of having a discriminatory attitude towards people living with HIV/AIDS were 61% (AOR = 0.39; 95%CI: 0.38, 0.40) lower among those individuals who had comprehensive knowledge towards HIV/AIDS as compared to those who had no comprehensive knowledge towards HIV/AIDS. Those individuals who were from the rural area had 33% (AOR = 0.67; 95%CI: 0.63, 0.72) lower odds of discriminatory attitude towards people living with HIV/AIDS as compared to their counterparts. Moreover, being from the Central and Eastern SSA region were 47% (AOR = 0.53; 95%CI: 0.50, 0.56) and 36% (AOR = 0.64; 95%CI: 0.61, 0.68) lower odds of having a discriminatory attitude towards people living with HIV/ AIDS, respectively, as compared to those from Western SSA region (Table 3).

**Random effect analysis.** As shown in Table 4, the ICC value in model 1 indicates that about 21.1% of the variability of discriminatory attitude towards people living with HIV/AIDS was attributable due to the difference between communities/clusters. Besides, the highest PCV value in model 3 revealed that 48.32% of the variability of discriminatory attitude towards people living with HIV/AIDS was explained by both individual and community-level variables.

**Table 3. Multilevel analysis for assessing factors associated with discriminatory attitude towards people living with HIV/AIDS in sub-Saharan Africa.**

| Variables | Model 1 | Model 2 | Model 3 | Model 4 |
|---|---|---|---|---|
| Sex | | | | |
| Male | | 1.00 | | 1.00 |
| Female | | 1.05 (1.02, 1.088 | | 1.02 (0.99, 1.06) |
| Age | | | | |
| 15–19 | | 1.00 | | 1.00 |
| 20–24 | | 0.81 (0.78, 0.84) | | 0.80 (0.78, 0.83) |
| 25–29 | | 0.77 (0.74, 0.80) | | 0.76 (0.73, 0.79) |
| 30–34 | | 0.68 (0.65, 0.70) | | 0.67 (0.64, 0.70) |
| 35–39 | | 0.69 (0.65, 0.71) | | 0.67 (0.64, 0.70) |
| 40–44 | | 0.63 (0.60, 0.66) | | 0.63 (0.60, 0.66) |
| 45–49 | | 0.66 (0.62, 0.69) | | 0.65 (0.62, 0.68) |
| Educational status | | | | |
| No education | | 1.00 | | 1.00 |
| Primary | | 0.40 (0.39, 0.42) | | 0.44 (0.42, 0.46) |
| Secondary | | 0.36 (0.35, 0.38) | | 0.37 (0.36, 0.39) |
| Higher | | 0.27 (0.26, 0.29) | | 0.27 (0.26, 0.29) |
| Occupation | | | | |
| Not working | | 1.00 | | 1.00 |
| Working | | 1.02(0.99, 1.05) | | 1.02 (0.99, 1.05) |
| Marital status | | | | |
| Single | | 1.00 | | 1.00 |
| Married | | 0.94 (0.90, 0.97) | | 0.93 (0.90, 0.96) |
| Widowed/separated/divorced | | 0.69(0.66, 0.73) | | 0.70 (0.66, 0.74) |
| Wealth status | | | | |
| Poorest | | 1.00 | | 1.00 |
| Poorer | | 0.92 (0.88, 0.96) | | 0.90 (0.87, 0.95) |
| Middle | | 0.83 (0.79, 0.87) | | 0.78 (0.74, 0.82) |
| Richer | | 0.75 (0.71, 0.80) | | 0.65 (0.61, 0.69) |
| Richest | | 0.63 (0.58, 0.67) | | 0.49 (0.46, 0.53) |
| Sex of household head | | | | |
| Male | | 1.00 | | 1.00 |
| Female | | 0.89 (0.86, 0.92) | | 0.90 (0.87–0.92) |
| Contraceptive use | | | | |
| No | | 1.00 | | 1.00 |
| Yes | | 0.72 (0.70, 0.74) | | 0.74 (0.72, 0.76) |
| Media exposure | | | | |
| No | | 1.00 | | 1.00 |
| Yes | | 1.01 (0.98, 1.04) | | 0.95 (0.92, 0.98) |
| Comprehensive knowledge of HIV/AIDS | | | | |
| No | | 1.00 | | 1.00 |
| Yes | | 0.38 (0.37, 0.39) | | 0.39 (0.38, 0.40) |
| Residence | | | | |
| Urban | | | 1.00 | 1.00 |
| Rural | | | 1.30 (1.21, 1.39) | 0.67 (0.63, 0.72) |
| Region of Africa | | | | |
| West Africa | | | 1.00 | 1.00 |
| Central Africa | | | 0.50 (0.46, 0.55) | 0.53 (0.50, 0.56) |

(*Continued*)

**Table 3.** (Continued)

| Variables | Model 1 | Model 2 | Model 3 | Model 4 |
|---|---|---|---|---|
| East Africa | | | 0.49 (0.46, 0.52) | 0.64 (0.61, 0.68) |
| Community-level of women literacy | | | | |
| Low | | | 1.00 | 1.00 |
| High | | | 1.01(0.85, 1.19) | 0.84 (0.75, 1.01) |
| Community-level media exposure | | | | |
| Low | | | 1.00 | 1.00 |
| High | | | 0.99 (0.84, 1.16) | 1.06 (0.91, 1.24) |

Moreover, the lowest deviance, which was 382,838.48, in model 4 revealed that model 4 was the best-fitted model for the data (Table 4).

## Discussion

This study aimed to assess discriminatory attitude towards people living with HIV/AIDS and its associated factors in 15 sub-Saharan African nations. In this study, the prevalence of discriminatory attitude towards people living with HIV/AIDS was 47.08% (95% CI: 47.08, 47.42). This finding is lower than a study conducted in Ethiopia and Nigeria [17, 18] and higher than a study finding in Pakistan [23]. This discrepancy may be due to the difference in the study population (since this study incorporates study participants in SSA), sociocultural and socio-economic differences between countries, as well as due to the difference in the study period and sample size (this study was based on pooled analysis).

This study also identified the individual and community level variables that were associated with discriminatory attitude towards people living with HIV/AIDS. The odds of having a discriminatory attitude towards people living with HIV/AIDS was higher among younger adults. This is congruent with studies conducted in Botswana [19]. This could be because people of a younger age are more reliant on their families and are less likely to get HIV/AIDS information. Furthermore, this group of people may not have the opportunity to get health education at the workplace or through other experiences that older people may have. Being not having formal education was associated with higher odds of discriminatory attitude towards people living with HIV/AIDS. This finding coincides with other study findings in Ethiopia [17, 28], Pakistan [23, 27], and Tajikistan [26]. The possible explanation is being educated enhances knowledge of a person in general and their exposure to modern media and modern health facilities. Besides, it may be since education is a powerful tool that affects the attitudes of individuals by promoting a better understanding of HIV/AIDS. Furthermore, educated individuals are more generous in accepting people living with HIV/AIDS and in showing readiness to respect the

**Table 4. Community-level variability (random effect analysis) of discriminatory attitude towards people living with HIV/AIDS and model fitness.**

| Parameter | Model 1 | Model 2 | Model 3 | Model 4 |
|---|---|---|---|---|
| Community variability (SE) | 0.880 (0.050) | 0.515 (0.035) | 0.647 (0.046) | 0.446 (0.033) |
| ICC (%) | 21.10 | 13.53 | 16.43 | 11.94 |
| PCV (%) | Reference | 41.48 | 26.48 | 49.32 |
| Model fitness | | | | |
| Deviance | 429,503.06 | 387,376.12 | 420,875.86 | 382,838.48 |

Note: SE = Standard Error, ICC = Intraclass Correlation Coefficient, PCV = Proportionate Change in Variance.

survival and interactional rights of the victims [30]. Moreover, education accelerates favorable socio-cultural change and helps individuals challenge misconceptions and traditional beliefs associated with the epidemic and people living with HIV/AIDS [31].

Household wealth index was another factor that was associated with discriminatory attitude towards people living with HIV/AIDS. The odds of having a discriminatory attitude towards people living with HIV/AIDS was higher among those adults who were from households with low socioeconomic/wealth status. This is in agreement with studies conducted elsewhere [23, 26, 27]. This could be because people from a higher socioeconomic background had better and more relevant knowledge, maybe more educated, may have better access to media, and are more protective and conscious of their health problems. In addition, in this study, being married or having a history of marriage had lower odds of having a discriminatory attitude towards people living with HIV/AIDS as compared to those who were never married/single. This is contrary to studies conducted in Ethiopia and Nigeria, which revealed that married individuals have a higher discriminatory attitude toward people living with HIV/AIDS than singles [18, 21]. The possible reason why married/those having marriage history had a higher discriminatory attitude towards people living with HIV/AIDS, in this study, may be due to gaining knowledge through discussion with their partner or since this group of individuals might be older and have a chance of gaining knowledge regarding HIV/AIDS through their experience.

The study at hand also revealed that exposure to media was associated with lower odds of discriminatory attitude towards people living with HIV/AIDS. This is in agreement with studies conducted in Ethiopia [17, 28] and Pakistan [27]. This could be because successful media communication raises an individual's knowledge of HIV/AIDS. In addition, the media disseminates accurate information that dismisses pre-existing myths and harmful attitudes towards HIV/AIDS. Besides, media services may assist individuals in learning more from others' experiences and improving their perceptions of the condition and people affected by HIV/AIDS. Furthermore, the media can affect people's attitudes and behavior regarding HIV/AIDS through boosting awareness and fostering positive attitudes, as well as fostering value systems that favor kindness and care for HIV/AIDS victims.

In addition, being having comprehensive knowledge of HIV/AIDS was associated with lower odds of discriminatory attitude towards people living with HIV/AIDS. This is in concordance with different studies conducted elsewhere [17, 22–26]. The most obvious explanation is that having accurate information regarding HIV transmission methods and the myths connected with AIDS transmission reduces stigmatizing behaviors and discriminatory attitudes toward HIV-positive people. Moreover, the current study also revealed that being from female head households and being using any contraceptive method was associated with lower odds of discriminatory attitude towards people living with HIV/AIDS.

The study at hand also revealed that community-level factors such as residence and SSA region were associated with discriminatory attitude towards people living with HIV/AIDS. Consistent with studies conducted elsewhere [21, 23, 28]; this study also identified regional variations in terms of discriminatory attitude towards people living with HIV/AIDS. This may be due to the different traditional behaviors and conservative outlooks that aggravate discrimination in regions of SSA Africa. Surprisingly and contrary to different study findings [21, 27], the current study revealed that being living in a rural area was associated with lower odds of discriminatory attitude towards people living with HIV/AIDS as compared to those who were from urban areas. This may be due to the advancement of the extension program in remote and rural areas. However, the authors recommend further investigation in this regard.

This study was based on representative data from SSA and with suitable statistical analysis (multilevel analysis). As a result, policymakers, as well as governmental and non-governmental

groups, can use it to make relevant actions. The study, however, had limitations because it was based on the information provided in the survey data and, therefore, we cannot incorporate the important variables such as cultural norms and perceptions about people living with HIV/AIDS. Besides, existing data sets only allows use of variables measured in the existing data set, as it is a case for the outcome variable that is measured by only two questions. Furthermore, because it was based on survey data, the cause and effect relationship between the outcome variable and independent variables cannot be demonstrated. As a result, caution is advised when interpreting the study's findings.

## Conclusion

The prevalence of discriminatory attitude towards people living with HIV/AIDS in SSA was high. Both individual and community-level factors were associated with discriminatory attitude towards people living with HIV/AIDS. Therefore, special attention should be given to those who are poor, uneducated, and younger adults. In addition, it is better to strengthen the accessibilities of different media for adult populations to create an appropriate attitude towards people with HIV/AIDS.

## Supporting information

**S1 Table. Sociodemographic characteristics of respondents by regions of Africa.**
(DOCX)

**S2 Table. Multilevel analysis for assessing factors associated with discriminatory attitude towards people living with HIV/AIDS among reproductive age women and men, analyzed separately.**
(DOCX)

**S3 Table. Multilevel analysis for assessing factors associated with discriminatory attitude towards people living with HIV/AIDS by regions of sub-Saharan Africa.**
(DOCX)

## Acknowledgments

Our deepest gratitude and appreciation go to the measure DHS program for allowing us to use the data set.

## Author Contributions

**Conceptualization:** Achamyeleh Birhanu Teshale, Getayeneh Antehunegn Tesema.

**Data curation:** Achamyeleh Birhanu Teshale, Getayeneh Antehunegn Tesema.

**Formal analysis:** Achamyeleh Birhanu Teshale, Getayeneh Antehunegn Tesema.

**Investigation:** Achamyeleh Birhanu Teshale, Getayeneh Antehunegn Tesema.

**Methodology:** Achamyeleh Birhanu Teshale, Getayeneh Antehunegn Tesema.

**Resources:** Achamyeleh Birhanu Teshale.

**Software:** Achamyeleh Birhanu Teshale, Getayeneh Antehunegn Tesema.

**Validation:** Achamyeleh Birhanu Teshale, Getayeneh Antehunegn Tesema.

**Visualization:** Achamyeleh Birhanu Teshale, Getayeneh Antehunegn Tesema.

**Writing – original draft:** Achamyeleh Birhanu Teshale, Getayeneh Antehunegn Tesema.

**Writing – review & editing:** Achamyeleh Birhanu Teshale, Getayeneh Antehunegn Tesema.

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
