## [Decision Letter · Decision Letter 0]

4 Sep 2021

PONE-D-21-17959

Discriminatory attitude towards people living with HIV/AIDS and its associated factors among adult population in Sub-Saharan Africa.

PLOS ONE

Dear Dr. Teshale,

Thank you for submitting your manuscript to PLOS ONE. After careful consideration, we feel that it has merit but does not fully meet PLOS ONE’s publication criteria as it currently stands. Therefore, we invite you to submit a revised version of the manuscript that addresses the points raised during the review process.

We look forward to receiving your revised manuscript.

Kind regards,

Orvalho Augusto, MD, MPH

Academic Editor

PLOS ONE

Journal Requirements:

Additional Editor Comments (if provided):

This is an important contribution to raise the awareness of discriminatory attitudes towards people living with HIV/AIDS. The authors used DHS datasets from 15 countries and conducted two analyses. One for the prevalence of discrimination. And to study the factors they used multilevel regression analysis to identify potential determinants. However, a few issues.

I. Major

1. DHS is designed to collect mother and child information. The HIV information is either secondary or additional on these surveys. This may make the woman selected to be likely to have had a recent pregnancy (or married) when compared to another woman in the community. How this was accounted for here? And why not do an additional analysis with males and females separated?

2. Moreover to the previous point, this analysis disregards that the SSA is not one big Africa. Another analysis that looks at least regionally would be appropriate here.

3. Discrimination is a bit hard to measure. The two questions used to assess this may not perform the same way across the countries included. This must be discussed.

4. It is to cause concern that the Southern Africa region is not included in the analysis when there are countries for that region (Malawi, Zimbabwe and Zambia). What classification did the authors use?

5. Statistical analysis. Why use deviance to decide which model to choose? The deviance does not penalize the addition of more variables. BIC or AIC would be better. Anyways, I would keep model 4 by pre-specification (and the analysis has been done).

II. Minor:

Abstract

1. Please add the unweighted numbers to the results.

Background

No comments

Methods

1. Line 97 - please list the countries included together with the name of the region

2. Line 100 - please add the unweighted numbers

3. Lines 120 to 123 are a repetition of what is stated between lines 136 to 146.

4. Lines 145 please cite Stata properly

5. For the prevalence calculation how the confidence intervals were computed

6. Did you incorporate the weights in the multilevel models?

Results

1. Table 1 please add the unweighted counts.

2. Table 2 please do this table per country (or region) and add such a table to the supplements.

3. Table 3 - in the supplements add one table for the females and another for males. As well there should by region.

4. Figure 2 - There is some strange grouping of the regions. Why not the Southern region?

5. Lines 199 - The fixed effects analysis is not documented in the statistical analysis. Can you describe what is done in this model?

Discussion

1. Line 320 - what “author”? Shouldn’t be “authors”?

2. Please expand the discussion of the limitations in light of the issues raised above

Reviewers' comments:

Reviewer's Responses to Questions

**Comments to the Author**

1. Is the manuscript technically sound, and do the data support the conclusions?

Reviewer #1: No

Reviewer #2: Partly

2. Has the statistical analysis been performed appropriately and rigorously? 

Reviewer #1: Yes

Reviewer #2: No

3. Have the authors made all data underlying the findings in their manuscript fully available?

Reviewer #1: Yes

Reviewer #2: Yes

4. Is the manuscript presented in an intelligible fashion and written in standard English?

Reviewer #1: No

Reviewer #2: Yes

5. Review Comments to the Author

Reviewer #1: Suggest title revision: Discriminatory attitude .... in 15 sub-Saharan African nations (there are a total of some 50)

Add section on limitations, i.e. existing data sets only allows use of variable measured in the existing data set, particularly a potential issue with the dichotomous dependent variable.

Expand implications for HIV work, i.e. targeting identify sub-groups - conclusion is very brief.

Reviewer #2: Thank you for your efforts in writing this manuscript.

Generally, HIV-related discrimination remains high but varies among countries. As of 2019 Guinea had about 80% and South Africa about 16.9% of people with discriminatory attitudes towards people living with HIV/AIDS. Is there a reason why South Africa, with the highest number of people living with HIV/AIDS is not included in your study despite the availability of data on South Africa from the Population-based Survey 2014-2018? What has South Africans done differently to reduce discriminatory attitudes as compared to Guinea and other countries in your study? Data from South Africa is pertinent and needs to be discussed in your manuscript.

Please check the following lines:

87 peoples?

99 (why women in brackets) Why?

299 peoples?

325 (residual confounders is there) Expain what you mean here.

Page 36: Prevalence...by individual countries

It should read country and not 'conutry'

Proofreading is necessary.

6. PLOS authors have the option to publish the peer review history of their article (what does this mean?). If published, this will include your full peer review and any attached files.

Reviewer #1: **Yes: **William L Holzemer

Reviewer #2: No

---

## [Author Response · Author response to Decision Letter 0]

19 Oct 2021

October18, 2021 

Authors’ response to editor’s and reviewers comments

Title: Discriminatory attitude towards people living with HIV/AIDS and its associated factors among adult population in Sub-Saharan Africa.

Manuscript number: PONE-D-21-17959

Dear all thank you for your constructive comments for the betterment of our manuscript. Below is the point-by-point response for issues you raised. In addition, we have amended our manuscript based on your comments, suggestions, and journals guideline. 

Response to Editor comments

1. DHS is designed to collect mother and child information. The HIV information is either secondary or additional on these surveys. This may make the woman selected to be likely to have had a recent pregnancy (or married) when compared to another woman in the community. How this was accounted for here? And why not do an additional analysis with males and females separated?

Author’s response: Thank you for the comment. In the DHS survey all women age 15-49 and all men age 15-59 who were either permanent residents of the selected households or visitors who stayed in the household the night before the survey were eligible to be interviewed.

Our main intention here, in this analysis, is to assess discriminatory attitude and its associated factors among adult populations. There are many individual studies that are conducted to this problem among males and females separately. Therefore, we have conducted this study whether the problem (discriminatory attitude) is different between males and females using sex as one of the factor.

In the DHS data there is HIV information. However, due to the secondary nature of the data, it may not contain all information and this is acknowledged in the discussion section. 

2. Moreover to the previous point, this analysis disregards that the SSA is not one big Africa. Another analysis that looks at least regionally would be appropriate here.

Author’s response: Thank you. Based on your recommendation, we have conducted analysis regionally and put this analysis as supplementary file (see S3 Table). 

3. Discrimination is a bit hard to measure. The two questions used to assess this may not perform the same way across the countries included. This must be discussed.

Author’s response: Dear Editor, Thank you for raising an important issue. In the DHS survey, the two questions were used to assess the discriminatory attitude towards HIV/AIDS and these were collected in similar way in all DHS surveys, except in some country surveys in which there is no collected information about discriminatory attitude (we have excluded such surveys in this study). However, measuring discriminatory attitude using these two questions is not enough and this is acknowledged, in the revised manuscript, as limitation in the discussion section. 

4. It is to cause concern that the Southern Africa region is not included in the analysis when there are countries for that region (Malawi, Zimbabwe and Zambia). What classification did the authors use?

Author’s response: Dear Editor, thank you. There are four African regions namely, Eastern, western, southern, and central. However, for this study, there was no countries in southern region with full information about discriminatory attitude. Therefore, we have assessed discriminatory attitude in three regions of Africa, particularly regions in sub-Saharan Africa. The above-mentioned countries (Malawi, Zimbabwe and Zambia) are under eastern Africa. As you know there are different classifications of African regions, however, many studies such as Tessema ZT et al, 2020 also considers these countries as Eastern Africa. In addition, when we access the DHS data these countries are under East African countries. 

5. Statistical analysis. Why use deviance to decide which model to choose? The deviance does not penalize the addition of more variables. BIC or AIC would be better. Anyways, I would keep model 4 by pre-specification (and the analysis has been done).

Author’s response: Thank you. We have used Deviance to choose our model since the models were nested. 

6. Abstract

 Please add the unweighted numbers to the results.

Author’s response: Thank you. We have added the unweighted numbers to the results

7. Methods

1. Line 97 - please list the countries included together with the name of the region

Author’s response: Thank you for the important comment you raised. However, this statement is re arranged and we only put list of countries we have used for the final analysis with their respective region in the revised manuscript (see Table 1).

2. Line 100 - please add the unweighted numbers

Author’s response: Thank you. We have added the unweighted numbers.

3. Lines 120 to 123 are a repetition of what is stated between lines 136 to 146.

Author’s response: We have considered your comment and removed the statements/phrases presented in line 120 to 123 in the revised manuscript. 

4. Lines 145 please cite Stata properly

Author’s response: Thank you. We have considered the comment in the revised manuscript.

5. For the prevalence calculation how the confidence intervals were computed

Author’s response: We have calculated the confidence interval for the prevalence using “prop outcome variable” stata command. 

6. Did you incorporate the weights in the multilevel models?

Author’s response: Yes, the multilevel models was all weighted. 

8. Results

1. Table 1 please add the unweighted counts.

Author’s response: Thank you. Based on your comment, we have added the unweighted counts in Table 1.

2. Table 2 please do this table per country (or region) and add such a table to the supplements.

Author’s response: We have considered your comment in the revised manuscript (see S1 Table).

3. Table 3 - in the supplements add one table for the females and another for males. As well there should by region.

Author’s response: Thank you. We have incorporated the multilevel analysis result for males and females, as well as per region as supplementary file (See S2 Table and S3 Table). 

4. Figure 2 - There is some strange grouping of the regions. Why not the Southern region?

Author’s response: Thank you. There was no countries with recorded information about discriminatory attitude in countries from southern African region and that is why we did not incorporate southern African region in the whole analysis. 

5. Lines 199 - The fixed effects analysis is not documented in the statistical analysis. Can you describe what is done in this model?

Author’s response: Thank you for your comment. The fixed effect analysis means simply the analysis conducted to assess factors associated with discriminatory attitude. For your information, table 3 and its description in the fixed effects analysis section is in general the fixed effect analysis result. 

9. Discussion

1. Line 320 - what “author”? Shouldn’t be “authors”?

Author’s response: Considered in the revised manuscript.

2. Please expand the discussion of the limitations in light of the issues raised above

Author’s response: Thank you we have revised the discussion of the limitation based on your comment and reviewers suggestion. 

Response to Reviewer #1 comments: 

1. Suggest title revision: Discriminatory attitude .... in 15 sub-Saharan African nations (there are a total of some 50)

Author’s response: Thank you for your comment. We have adjust our title according to your comment.

Add section on limitations, i.e. existing data sets only allows use of variable measured in the existing data set, particularly a potential issue with the dichotomous dependent variable.

Expand implications for HIV work, i.e. targeting identify sub-groups - conclusion is very brief.

 Author’s response: Thank you for the important issue raised. We have incorporated your suggestion in the revised manuscript. 

Response to Reviewer #2 comments:

1. Generally, HIV-related discrimination remains high but varies among countries. As of 2019 Guinea had about 80% and South Africa about 16.9% of people with discriminatory attitudes towards people living with HIV/AIDS. Is there a reason why South Africa, with the highest number of people living with HIV/AIDS is not included in your study despite the availability of data on South Africa from the Population-based Survey 2014-2018? What has South Africans done differently to reduce discriminatory attitudes as compared to Guinea and other countries in your study? Data from South Africa is pertinent and needs to be discussed in your manuscript.

Author’s response: Thank you for the important comment. Even though south Africa had the most recent DHS survey conducted between 2015 to 2020, it had no recorded information about discriminatory attitude (i.e variables v825 and v857a had no observation).

2. Please check the following lines:

87 peoples?

99 (why women in brackets) Why?

299 peoples?

325 (residual confounders is there) Expain what you mean here.

Page 36: Prevalence...by individual countries

It should read country and not 'conutry'

Proofreading is necessary.

Author’s response: Thank you. We have considered these comments in the revised manuscript.

---

## [Decision Letter · Decision Letter 1]

15 Dec 2021

Discriminatory attitude towards people living with HIV/AIDS and its associated factors among adult population in 15 sub-Saharan African nations.

PONE-D-21-17959R1

Dear Dr. Teshale,

We’re pleased to inform you that your manuscript has been judged scientifically suitable for publication and will be formally accepted for publication once it meets all outstanding technical requirements.

Kind regards,

Orvalho Augusto, MD, MPH

Academic Editor

PLOS ONE

Additional Editor Comments (optional):

This is the second revision of this manuscript The authors responded fully to the reviewers' comments.

Few more issues:

1. Please correct STATA to Stata. Stata is not an acronym (see the official Stata documentation).

2. I am OK with (or not) the use of weighted multilevel analysis. In my response, the authors said that they did use weighted multilevel analysis. I would ask them to make this clear in the text as well. The use of weights here could be a bit problematic as the weights from each survey could be standardized (or not) and therefore lead to questions like whether these weights are comparable. So, some may see issues. Could you add the unweighted multilevel analysis in the supplementary materials as well? (for the same models in table 3)

3. Please make sure that fractional numbers (in proportions/fractions) have 1 or 2 (choose one) decimal places and stick to this. It is not OK that in the very same paragraph (lines 175 to 181) some have 2 decimals and other (the 41%) none decimal places.

4. Line 177 - better say something like “better of” rather than “richest households”.

5. Table 2 - please state that the frequencies are weighted.

6. How the prevalence for the regions was computed (figure 2)? Would be good to use a meta-synthesis for this rather than simple weighted prevalence.

7. Tables in the supplementary materials have **** or something similar, can you put below the table what those mean?

8. Line 247 the 382838.4 corresponds to model 4 not to model 3.

Reviewers' comments:

Reviewer's Responses to Questions

**Comments to the Author**

1. If the authors have adequately addressed your comments raised in a previous round of review and you feel that this manuscript is now acceptable for publication, you may indicate that here to bypass the “Comments to the Author” section, enter your conflict of interest statement in the “Confidential to Editor” section, and submit your "Accept" recommendation.

Reviewer #2: (No Response)

2. Is the manuscript technically sound, and do the data support the conclusions?

Reviewer #2: Yes

3. Has the statistical analysis been performed appropriately and rigorously? 

Reviewer #2: Yes

4. Have the authors made all data underlying the findings in their manuscript fully available?

Reviewer #2: Yes

5. Is the manuscript presented in an intelligible fashion and written in standard English?

Reviewer #2: Yes

6. Review Comments to the Author

Reviewer #2: Thank you for considering some of my comments in revising your manuscript. However, your manuscript still needs to be proofread. There are some small errors that should be looked at. For example: lines 95-98; lines 99 and more in the text.

7. PLOS authors have the option to publish the peer review history of their article (what does this mean?). If published, this will include your full peer review and any attached files.

Reviewer #2: No

---

## [Editor Report · Acceptance letter]

28 Jan 2022

PONE-D-21-17959R1 

Discriminatory attitude towards people living with HIV/AIDS and its associated factors among adult population in 15 sub-Saharan African nations. 

Dear Dr. Teshale:

I'm pleased to inform you that your manuscript has been deemed suitable for publication in PLOS ONE. Congratulations! Your manuscript is now with our production department. 

Kind regards, 

on behalf of

Dr. Orvalho Augusto 

Academic Editor

PLOS ONE